# Comparing Rewinding and Fine-tuning in Neural Network Pruning Reproducibility Challenge 2021

## Reproduction Summary

**Scope of Reproducibility**

We are reproducing *Comparing Rewinding and Fine-tuning in Neural Networks*, by Renda et al. [2020]. In this work the authors compare three different approaches to retraining neural networks after pruning: 1) fine-tuning, 2) rewinding weights as in Frankle and Carbin [2019] and 3) a new, original method involving learning rate rewinding, building upon Frankle and Carbin [2019]. We reproduce the results of all three approaches, but we focus on verifying their approach, learning rate rewinding, since it is newly proposed and is described as a universal alternative to other methods.

We used CIFAR10 for most reproductions along with additional experiments on the larger CIFAR100, which extends the results originally provided by the authors. We have also extended the list of tested network architectures to include Wide ResNets (Zagoruyko and Komodakis [2016]). The new experiments led us to discover the limitations of learning rate rewinding which can worsen pruning results on large architectures.

**Methodology**

We implemented the code ourselves in Python with TensorFlow 2, basing our implementation of the paper alone and without consulting the source code provided by the authors. We ran two sets of experiments. In the reproduction set, we have striven to exactly reproduce the experimental conditions of Renda et al. [2020]. We have also conducted additional experiments, which use other network architectures, effectively showing results previously unreported by the authors. We did not cover all originally reported experiments – we covered as many as needed to state the validity of claims. We used Google Cloud resources and a local machine with 2x RTX 3080 GPUs.

**Results**

We were able to reproduce the exact results reported by the authors in all originally reported scenarios. However, extended results on larger Wide Residual Networks have demonstrated the limitations of the newly proposed learning rate rewinding – we observed a previously unreported accuracy degradation for low sparsity ranges. Nevertheless, the general conclusion of the paper still holds and was indeed reproduced.

**What was easy**

Re-implementation of the pruning and retraining methods was technically easy, as it is based on a popular and simple pruning criterion – magnitude pruning. Original work was descriptive enough to reproduce the results with satisfying results without consulting the code.

**What was difficult**

Not every design choice was mentioned in the paper, thus reproducing the exact results was rather difficult and required a meticulous choice of hyper-parameters. Experiments on ImageNet and WMT16 datasets were time consuming and required extensive resources, thus we did not verify them.

**Communication with original authors**

We did not consult the original authors, as there was no need to

# 1 Introduction

Neural network pruning is an algorithm leading to decrease the size of a network, usually by removing its connections or setting their weights to 0. This procedure generally allows obtaining smaller and more efficient models. It often turns out that these smaller networks are as accurate as their bigger counterparts or the accuracy loss is negligible. A common way to obtain such high quality sparse network is to prune it after the training has finished (Liu et al. [2019], Frankle and Carbin [2019]). Networks that have already converged are easier to prune than randomly initialized networks (Liu et al. [2019], Lee et al. [2018]). After pruning, more training is usually required to restore the lost accuracy. Although there are a few ways to retrain the network, finetuning might be the easiest and most often chosen by researchers and practitioners. (Liu et al. [2019], Renda et al. [2020]).

Lottery Ticket Hypothesis from Frankle and Carbin [2019] formulates a hypothesis that for every dense neural network, there exists a smaller subnetwork that matches or exceeds results of the original. The algorithm originally used to obtain examples of such networks is iterative magnitude pruning with weight rewinding, and it is one of the methods of retraining after pruning compared in this work.

# 2 Scope of reproducibility

Renda et al. [2020] formulated the following claims:

Claim 1: Widely used method of training after pruning: finetuning yields worse results than rewinding based methods (supported by figures 1, 2, 3, 4 and table 5)

Claim 2: Newly introduced learning rate rewinding works as good or better as weight rewinding in all scenarios (supported by figures 1, 2, 3, 4 and table 5, but not supported by figure 5)

Claim 3: Iterative pruning with learning rate rewinding matches state-of-the-art pruning methods (supported by figures 1, 2, 3, 4 and table 5, but not supported by figure 5)

# 3 Methodology

We aimed to compare three retraining approaches: 1) finetuning, 2) weight rewinding and 3) learning rate rewinding. Our general strategy that repeated across all experiments was as follows:

1. train a dense network to convergence,
2. prune the network using magnitude criterion: remove weights with smallest L1 norm,
3. retrain the network using selected retraining approach.

In the case of structured pruning: in step 2, we removed structures (rows or convolutional channels) with the smallest average L1 norm (Crowley et al. [2018]), rather than removing separate connections.

In the case of iterative pruning: the network in step 1 was not randomly initialized, but instead: weights from a model from a previous iterative pruning step were loaded as the starting point.

We trained all our networks using Stochastic Gradient Descent with Nesterov Momentum. The learning rate was decreased in a piecewise manner during the training, but momentum coefficient was constant and equal to $0.9$.

## 3.1 Model descriptions

In this report, we were focusing on an image recognition task using convolutional neural networks (LeCun [1988]). For most of our experiments, we chose to use identical architectures as Renda et al. [2020] to better validate their claims and double-check their results, rather than provide additional ones. Therefore, most of the used networks are residual networks, which were originally proposed in He et al. [2016a]. Additionally, to verify the general usefulness of pruning and retraining methods proposed in Renda et al. [2020] we extend the list of tested network architectures to much larger wide residual networks from Zagoruyko and Komodakis [2016].

### 3.1.1 Residual networks (ResNet)

Just as Renda et al. [2020], we chose to use the original version of ResNet as described in He et al. [2016a] rather than the more widely used, improved version (with preactivated blocks) from He et al. [2016b]. We created the models

ourselves, using TensorFlow (Abadi et al. [2015]) and Keras. We strove to replicate the exact architectures used by Renda et al. [2020] and He et al. [2016a] and train them from scratch.

| Model | Trainable parameters | Kernel parameters | CIFAR-10 | CIFAR-100 |
|---|---|---|---|---|
| ResNet-20 | 272 282 | 270 896 | 92.46% | – |
| ResNet-56 | 855 578 | 851 504 | 93.71% | 71.90% |
| ResNet-110 | 1 730 522 | 1 722 416 | 94.29% | 72.21% |

Table 1: ResNets architecture description, including baseline accuracy across datasets.

## Hyper-parameters

Learning rate started with $0.1$ and was multiplied by $0.1$ twice, after $36\,000$ and $54\,000$ iterations. One training cycle was $72\,000$ iterations in total. For all batch normalization layers, we set the batch norm decay to $0.997$, following Renda et al. [2020] which was also used in the original TensorFlow implementation[1]. We initialize network's weights with what is known as He uniform initialization from He et al. [2015]. We regularize ResNets, during both training and finetuning, using $L2$ penalty with $10^{-4}$ coefficient. In other words, the loss function (from which we calculate the gradients) looks as follows:

$$FinalLoss = CategoricalCrossentropy(GroundTruth, Prediction) + 10^{-4} \times \sum_{i \in W} w_i^2$$

### 3.1.2 Wide Residual Networks (Wide ResNet, WRN)

WRN networks were introduced in Zagoruyko and Komodakis [2016]. They are networks created by simply increasing the number of filters in preactivated ResNet networks (He et al. [2016b]).

| Model | Trainable parameters | Kernel parameters | CIFAR-10 |
|---|---|---|---|
| WRN-16-8 | 10 961 370 | 10 954 160 | 95.72% |

Table 2: Wide ResNet architecture description.

## Hyper-parameters

As Wide ResNets are newer and much larger than ResNets, hyper-parameters are slightly different. To choose them, we follow Zagoruyko and Komodakis [2016]. Learning rate starts with $0.1$ and multiplied by $0.2$ thrice: after $32\,000$, $48\,000$ and $64\,000$ iterations. Training lasts for $80\,000$ iterations. For all batch normalization layers, we use hyper-parameters from the newer TensorFlow implementation[2] with batch norm decay set to $0.9$. Following Zagoruyko and Komodakis [2016], we use larger $L2$ penalty for this network: $2 \times 10^{-4}$. Finally, the loss function is as follows:

$$FinalLoss = CategoricalCrossentropy(GroundTruth, Prediction) + 2 \times 10^{-4} \times \sum_{i \in W} w_i^2$$

## 3.2 Datasets

CIFAR-10 and CIFAR-100 are image classification datasets introduced in Krizhevsky et al.. Following Renda et al. [2020], we use all ($50\,000$) training examples to train the model.

---

[1]`https://github.com/tensorflow/models/blob/r1.13.0/official/resnet/resnet_model.py`
[2]`https://github.com/tensorflow/models/blob/r2.5.0/official/vision/image_classification/resnet/resnet_model.py`

| Dataset | Training examples | Validation examples | Classes | Resolution |
|---------|-------------------|---------------------|---------|------------|
| CIFAR-10 | 50 000 | 10 000 | 10 | 32×32 |
| CIFAR-100 | 50 000 | 10 000 | 100 | 32×32 |

Table 3: CIFAR datasets description.

### 3.2.1 Postprocessing

We used a standard postprocessing for both CIFAR-10 and CIFAR-100 datasets (Renda et al. [2020], Frankle and Carbin [2019], Zagoruyko and Komodakis [2016]). During training and just before passing data to the model, we:

1. standardized the input by subtracting the mean and dividing by the std of RGB channels (calculated on training dataset),
2. randomly flipped in horizontal axis,
3. added a four pixel reflection padding,
4. randomly cropped the image to its original size.

During the validation, we did only the first step of the above.

### 3.3 Experimental setup and code

Our ready-to-use code, which includes experiment definitions, can be found at `https://anonymous.4open. science/r/reproducing-comparing-rewinding-and-finetuning-1C5A`. It's written using TensorFlow (Abadi et al. [2015]) version 2.4.2 in Python. More details are included in the repository.

### 3.4 Computational requirements

Recreating the experiments required a modern GPU, training all models on CPU was virtually impossible. Training time varies depending on a lot of factors: network version and size, exact version of the deep learning library, and even the operating system. In our case, using TensorFlow 2.4.2 on Ubuntu and a single RTX 3080 GPU, the smallest of the used models, ResNet-20, takes about 20 minutes to train on CIFAR-10 dataset. To replicate our experiments, training at least a single baseline network and then, separately, a single pruned network, is required. To reduce computational requirements, we reused one dense baseline for multiple compression ratios. Approximated training time requirements can be seen in the table below.

| Model | Dataset | Number of iterations | Iterations per second | Time for training cycle |
|-------|---------|----------------------|-----------------------|-------------------------|
| ResNet-20 | CIFAR-10 | 72 000 | 59.0 | 22 min |
| ResNet-56 | CIFAR-10 | 72 000 | 28.6 | 43 min |
| ResNet-110 | CIFAR-10 | 72 000 | 15.9 | 77 min |
| WRN-16-8 | CIFAR-10 | 80 000 | 17.4 | 78 min |

Table 4: Time requirements for replicating or running experiments from this report. Reported times are obtained using a single RTX 3080 GPU in Linux environment, using TensorFlow in version 2.4.2.

For all our experiments in total, we used around 536 GPU hours.

## 4 Method description

We compare three methods of retraining after pruning. For all of them, the starting point is a network that was already trained to convergence, then pruned to a desired sparsity. The difference between the three retraining methods is what follows after it.

### 4.1 Fine-tuning

Fine-tuning is retraining with a small, constant learning rate – in our case, whenever fine-tuning was used, the learning rate was set to 0.001 as in Renda et al. [2020]. We finetune the network for the same number of iterations as the baseline – 72 000 iterations in the case of the original ResNet architecture. In this method, such long retraining would not be necessary in practical applications, since the network converges much faster.

### 4.2 Weight rewinding

Weight rewinding restores the network's weights from a previous point (possibly beginning) in the training history and then continues training from this point using the original training schedule – in our case a piecewise constant decaying learning rate schedule. When rewinding a network to iteration $K$ that originally trained for $N$ iterations: first prune the dense network that was trained for $N$ iterations. Then, for connections that survived, restore their values to $K$-th iteration from the training history. Then train to the convergence for the remaining $N - K$ iterations.

### 4.3 Learning rate rewinding

Learning rate rewinding continues training with weights that have already converged, but restores the learning rate schedule to the beginning, just as if we were training from scratch, and then trains to the convergence once again. This reminds the cyclical learning rates from Smith [2017]. Learning rate rewinding really is weight rewinding for $K = N$, but the final retraining is always for $N$ iterations.

## 5 Results

In most of our experiment, just as Renda et al. [2020], we investigate how does the trade-off between prediction accuracy and compression ratio look like. In one of the experiments (table 5) we verify only one compression ratio, but for the rest, we verify multiple. We report a median result out of 2 up to 12 trials for each compression ratio. To better utilize our compute capabilities, we decided to spend more training cycles in situations where there is no clear winner between the compared methods. On each plot, we include error bars showing 80% confidence intervals.

### 5.1 Results reproducing original paper

In this section, we include experiments that we successfully reproduced. They match the original ones within 1% error margin.

Across all scenarios where finetuning was tested, it was by far the worst of the three methods, which directly supports claim 1 (section 2). Weight rewinding and learning rate rewinding most often are equally matched, but in some cases learning rate rewinding works a little better.

**ResNets on CIFAR-10 dataset**

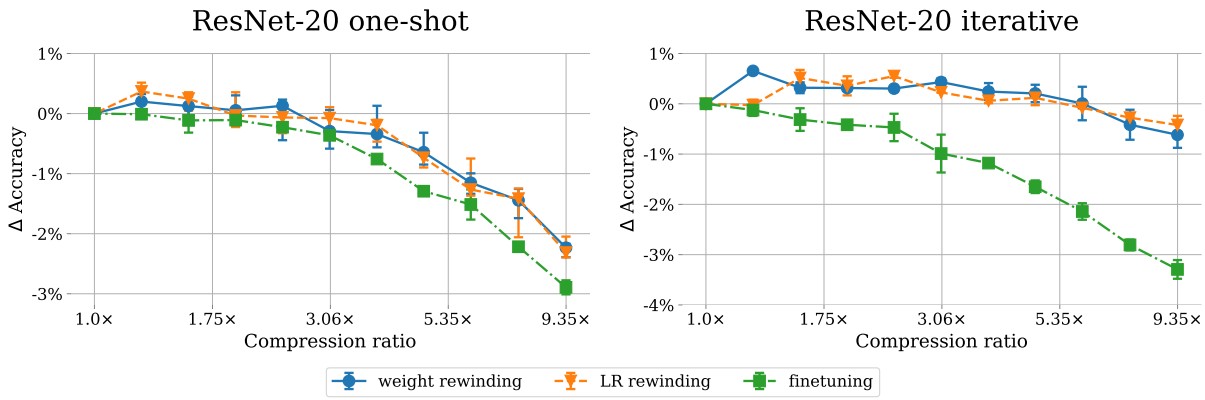

Figure 1: Results of ResNet-20 (table 1) on CIFAR-10 (table 3) with unstructured, magnitude pruning in versions: one-shot and iterative. Results show varying compression ratios. Maximal compression ratio (9.35×) means that there are only 29 000 non-zero kernel parameters left. This experiment supports claims 1, 2, 3 (section 2).

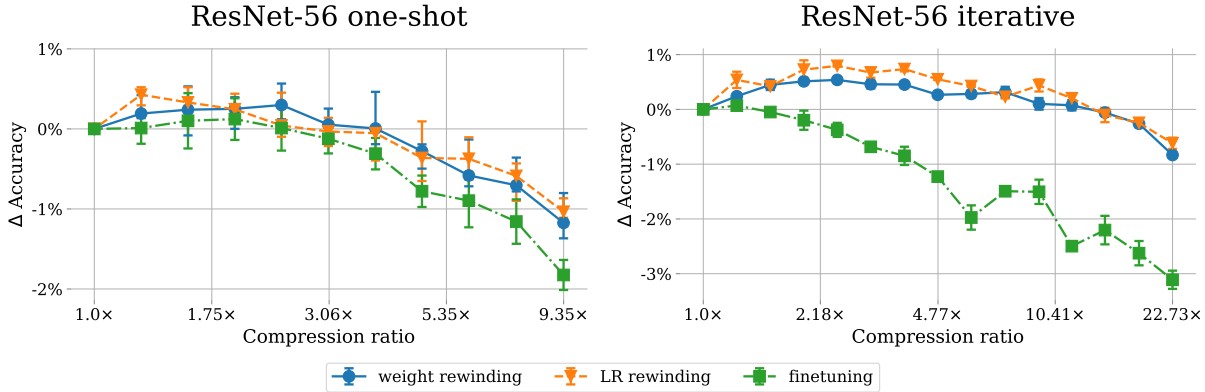

Figure 2: Results of ResNet-56 (table 1) on CIFAR-10 (table 3) with unstructured, magnitude pruning in versions: one-shot and iterative. Results with varying compression ratios. Maximal compression ratio means (22.73×) that there are only 37 600 non-zero kernel parameters left. This experiment supports claims 1, 2, 3 (section 2).

| Network | Dataset | Retraining | Sparsity | Test Accuracy |
|---------|---------|-----------|----------|---------------|
| ResNet-110 | CIFAR-10 | None | 0% | 94.29% |
| ResNet-110 | CIFAR-10 | LR rewinding | 89.3% | 93.74% |
| ResNet-110 | CIFAR-10 | weight rewinding | 89.3% | 93.73% |
| ResNet-110 | CIFAR-10 | finetuning | 89.3% | 93.32% |

Table 5: Results of ResNet-110 (table 1) trained on CIFAR-10 (table 3) with unstructured, one-shot magnitude pruning. Sparsity 89.3% corresponds to 9.35× compression ratio. This experiment supports claims 1, 2, 3 (section 2).

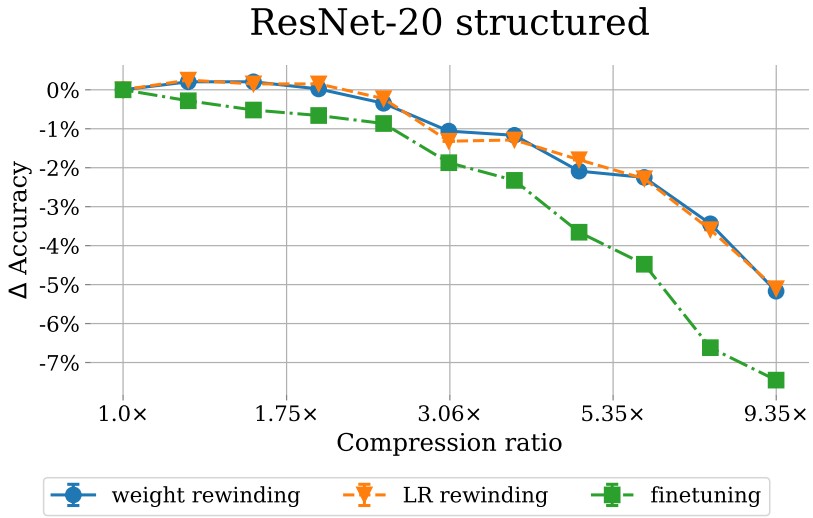

Figure 3: Results of ResNet-20 (table 1) on CIFAR-10 (table 3) with structured, one-shot, magnitude pruning. Results show varying compression ratios. Maximal compression ratio (9.35×) means that there are only 29 000 non-zero kernel parameters left in ResNet-20.

## 5.2 Results beyond original paper

**ResNets on CIFAR-100 dataset**

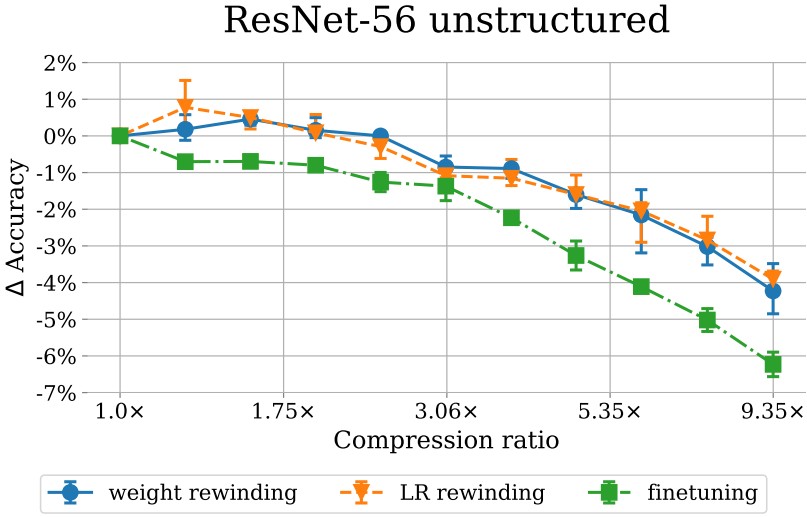

Figure 4: Results of ResNet-56 (table 1) on CIFAR-100 (table 3) with unstructured, one-shot, magnitude pruning. Results with varying compression ratios. Maximal compression ratio (9.35×) means that there are only 91 500 non-zero kernel parameters left. This experiment supports claims 1, 2, 3 (section 2) even though this scenario wasn't originally tested in Renda et al. [2020].

**WRN-16-8 on CIFAR-10 dataset**

 WRN-16-8 shows consistent behaviour – accuracy in the low sparsity regime is reduced in comparison to the baseline.
 In the case of iterative pruning, where each step is another pruning in the low sparsity regime, it leads to a large
 difference between the two retraining methods. Since for WRN-16-8 one-shot, low sparsity pruning shows a small
 regression in comparison to the baseline, this regression accumulates when pruning multiple times, as we do in iterative
 pruning. This can be seen in figure 5.

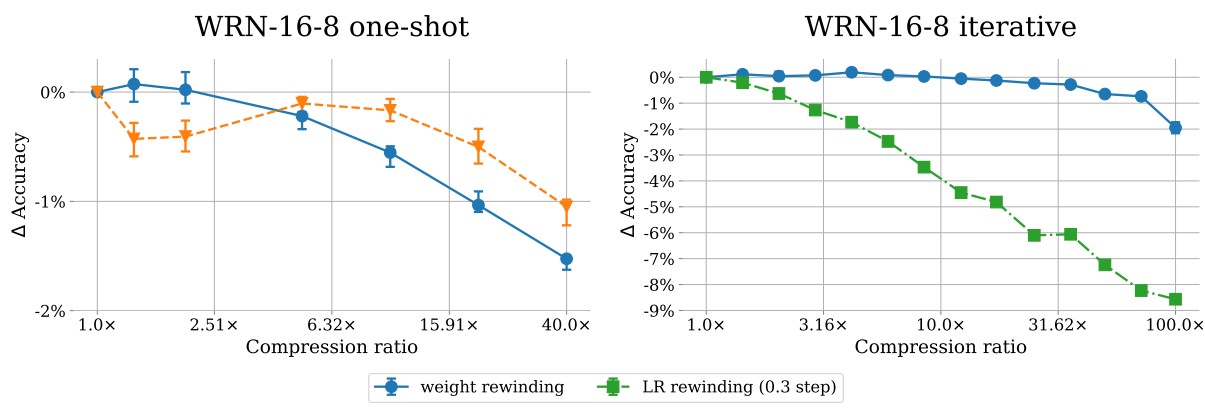

Figure 5: Results of WRN-16-8 (table 2) on CIFAR-10 (table 3) with unstructured, magnitude pruning in versions: one-shot and iterative. Results with varying compression ratios. Maximal compression ratio (100×) leaves 109 500 non-zero kernel parameters while achieving around 94% accuracy or around 95% when leaving 153 400 non-zero parameters. One can see catastrophic effects of low-sparsity pruning when using learning rate rewinding procedure.

For iterative pruning (figures 1, 2) we used a nonstandard step size of 30% per iterative pruning iteration, which was a way to reduce the computational requirements. We provide a comparison of our step size to the more commonly used 20%. We show that there is virtually no difference between both versions and the aforementioned catastrophic degradation occurs in both cases, as long as the step size is in the low sparsity regime.

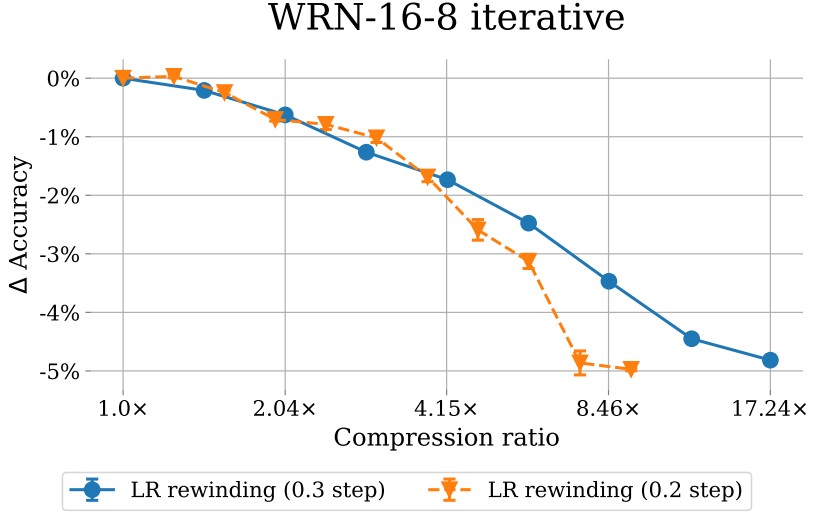

Figure 6: Results of WRN-16-8 (table 2) on CIFAR-10 (table 3) with unstructured, iterative, magnitude pruning with two different step sizes. Results show varying compression ratios and accuracy.

## 6 Discussion

We were able to confirm the general conclusion of Renda et al. [2020]. Fine-tuning can mostly be replaced by other retraining techniques, e.g., by weight rewinding as done by Frankle and Carbin [2019]. However, we have also shown in figure 5 that the newly proposed learning rate rewinding was a poor choice when we were pruning larger networks – in our case that was WRN-16-8. We believe this should be further examined as there might exist a simple workaround to this problem – a retraining procedure in between weight rewinding and learning rate rewinding which would work in all cases. Furthermore, it would be interesting to see where exactly learning rate rewinding starts losing accuracy in comparison to weight rewinding and why this catastrophic accuracy degradation occurs. Perhaps, the reason for it not occurring with the original ResNet architecture is the degree to which the larger networks overtrain – larger networks tend to overtrain more. Such an overtrained network might not be a good starting point for the retraining.

## Acknowledgements

The authors thank Polish National Science Center for funding under the OPUS-18 2019/35/B/ST6/04379 grant and the PlGrid consortium for computational resources.

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
