# OpenReview forum: "Comparing Rewinding and Fine-tuning in Neural Network Pruning"
_ML_Reproducibility_Challenge/2021/Fall — RC2021_

### Official Review · Reviewer_Yy1q · 2022-02-23
**Good reimplementation with additional results on wide ResNet**

**Rating:** 7
**Confidence:** 4

**Review:**

SUMMARY:

The authors of the report aim at reproducing the results of the paper "Comparing Rewinding and Fine-tuning in Neural Networks" by Renda et al. (2020). The original work discusses how weight rewinding, learning rate rewinding and finetuning perform on pruned neural networks. The methods are reimplemented from scratch and most of the claims of the original work were reproduced. Furthermore experiments were extended by using an additional wide ResNet.

STRENGTHS:
- Reproducibility Summary: there is a reproducibility summary.
- Code: The implementation used for the experiments was done from scratch using Tensorflow 2.0. The code has been submitted in an anonymized repository. There are some missing doc-strings, but overall the code is well-structured and for each experiment there exists a yaml-file containing the used hyperparameters.
- The authors discuss three claims from the original work and they were very clear about which results supports and which results do not support the claims of the original work.
- The used experimental setup is very nicely described. Hyperparameter settings are elaborated in a very detailed way and also the authors of the report were very clear about which model they used for which experiment.
- Results beyond the paper: There are results that go beyond the experimental evaluation of the original work. For one the authors evaluate unstructured pruning on the ResNet-56 trained on the CIFAR-100 dataset. Additionally, the authors analyze iterative and one-shot pruning on a wide residual network trained on the CIFAR-10 dataset.
- Communication with the original authors: no communication needed.

WEAKNESSES:
- The description of the methodology (Section 3) is somewhat unclear: For example, the formulation "remove weights with smallest L1 norm" is a little bit vague. It would be good if the authors could explain the difference between structured pruning and unstructured pruning at this point. Also to me it is not clear what is meant with "rows". Does this refer to rows of a filter? Also this section should mention the keyword "one-shot" to make it clear that this is the opposite of "iterative".
- l.58: what is a dense network? I assume that this refers to a neural network that was not pruned (instead to a neural network having dense layers)
- l.87: "Postprocessing" -> "Preprocessing and dataset augmentation"
- Figures: the ticks of the x-axis are very arbitrary. In my opinion, this is problematic because it makes reading off the results very difficult. It would be better to have a tick for each datapoint. Perhaps this could be realized by rotating the x-ticks by 90 degrees.

RECOMMENDATION:

The authors of the report did a good job at reproducing the results from the original paper. I would recommend to accept this submission into RC 2021 after the issues mentioned above are adequately addressed.

---

### Official Review · Reviewer_iGSg · 2022-03-01

**Rating:** 8
**Confidence:** 4

**Review:**

This report does an excellent job at a reproducing the original paper. I want to commend the authors at their outstanding tenacity to reproduce the results from the paper description alone. Only a few comments/suggestions for future reproduction efforts, but overall I think this is an excellent report and should definitely be accepted.

Suggestions:
- I was hoping there would be inline comments in the code you submitted, but couldn't find any. I think you do a good job making the code base easy to use and play with, but would suggest you comment important parts of the code. This would help with the extensibility of the ideas, and give better insight into the parts that weren't reported you worked hard to figure out.
- I appreciate the addition of CIFAR-100 as added dataset, but I do wonder if a dataset of a different type (i.e. not image classification) would have been better to address the robustness and generalizability of the original paper's methods.

---

### Official Review · Reviewer_R3Uo · 2022-03-06
**interesting results with encouragement to improve writing**

**Rating:** 7
**Confidence:** 4

**Review:**

Summary: This paper reproduces “Comparing Rewinding and Fine-tuning in Neural Networks”, by Renda et al. [2020]. In short, they measure the performance of three approaches for re-training pruned neural networks from pre-trained networks, including, fine-tuning, weight rewinding, and learning rewinding. The authors have been able to reproduce the results on the CIFAR10 and CIFAR100 datasets. However, the results regarding the large datasets (ImageNet and WMT16) have been excluded due to time and resource limitations that is understandable.

The authors have clearly stated the scope of reproducibility, and also described what was easy/difficult in reproducing the results. They implemented the code from scratch and did not use the original code for the paper. The authors did not contact the original authors. No ablation studies were performed.

Pros

•	Authors have extended the results; they measured the performance of the method on Wide ResNet for various compression ratios which was not considered in the original paper.

•	Authors have gained new observations of the proposed method by Renda et. Al. [2020]. Learning rate rewinding has a low performance in retraining large pruned networks (here Wide ResNet) particularly for high pruning rate.

Cons

•	Writing style and use of English in the paper can by significantly improved.

•	The description of the obtained results in Section 5.1 is limited and can be expanded.



Minor comments

•	In Figure 5, legends for the plots is not complete (orange line).

•	(Page 2 – line 22) + (Figure 5 caption) + … :  Please double-check in the paper: Do you mean “high” sparsity range instead of “low”? When the pruning/compression rate is increased, the sparsity increases and density decreases

Overall, I believe the obtained results and discussions can be interesting for the community and therefore, I recommend acceptance.

---

### Meta-Review · Program_Chairs · 2022-04-09

**Recommendation:** Accept
**Confidence:** 5

**Metareview:**

A solid contribution to the reproducibility challenge.  The work is accepted.

---

### Decision · Program_Chairs · 2022-04-09

**Decision:**

Accept

**Comment:**

Following the recommendation of reviewers and meta-reviewer, the paper is accepted for ML Reproducibility Challenge 2021, and will be published in the upcoming special edition of ReScience Journal.